# Dual function of Rab1A in secretion and autophagy: hypervariable domain dependence

Valeriya Gyurkovska, Rakhilya Murtazina ⓘ, Sarah F Zhao ⓘ, Sojin Shikano, Yukari Okamoto, Nava Segev ⓘ

We currently understand how the different intracellular pathways, secretion, endocytosis, and autophagy are regulated by small GTPases. In contrast, it is unclear how these pathways are coordinated to ensure efficient cellular response to stress. Rab GTPases localize to specific organelles through their hypervariable domain (HVD) to regulate discrete steps of individual pathways. Here, we explored the dual role of Rab1A/B (92% identity) in secretion and autophagy. We show that although either Rab1A or Rab1B is required for secretion, Rab1A, but not Rab1B, localizes to autophagosomes and is required early in stress-induced autophagy. Moreover, replacing the HVD of Rab1B with that of Rab1A enables Rab1B to localize to autophagosomes and regulate autophagy. Therefore, Rab1A-HVD is required for the dual functionality of a single Rab in two different pathways: secretion and autophagy. In addition to this mechanistic insight, these findings are relevant to human health because both the pathways and Rab1A/B were implicated in diseases ranging from cancer to neurodegeneration.

## Introduction

In the intracellular trafficking pathways, proteins are transported between intracellular organelles. While flow from the inside of the cells to the plasma membrane (PM) occurs through the secretory pathway, traffic from the PM to the lysosome, the cellular degradative compartment, happens through the endocytic pathway. Another pathway that leads to the lysosome is macro-autophagy (will use autophagy henceforth). Autophagy is a constitutive recycling pathway that can be further induced by different kinds of stress. In mammalian tissue culture cells, growth factors deprivation stimulates stress-induced autophagy through which most cellular components can be degraded for the reuse of their building blocks. In this pathway, a conserved complex of Atg proteins is required to form a double-membrane organelle termed autophagosome (AP) that engulfs the cargo to be delivered for degradation in the lysosome (Feng et al, 2014; Ohsumi, 2014).

The conserved family of Rab GTPases, which includes ~70 human members, regulates all intracellular traffic pathways in eukaryotic cells: secretion, endocytosis, and autophagy (Homma et al, 2021). Rabs localize to specific compartments where they are activated by their guanine nucleotide exchange factors. On these membranes, they recruit their downstream effectors to organize membrane micro-domains that mediate vesicular trafficking (Segev, 2001; Zerial & McBride, 2001). Although all Rabs share signature similarities, they are most different in their C-terminal hypervariable domain (HVD; ~25 amino acids). This domain allows individual Rabs to attach to the specific membranes on which they function (Li et al, 2014).

The yeast Ypt1 is required for execution of early steps in the secretory and autophagic pathways (Lipatova et al, 2015). In autophagy, Ypt1 GTPase, together with the Atg complex, is necessary for the first step of autophagy, the formation of the pre-autophagosomal structure (Lipatova et al, 2012; Lipatova & Segev, 2012). Ypt1 has two human homologs: Rab1A and Rab1B that share 92% identity at the amino acid sequence (see Fig 8A). For comparison, the level of identity between Rab1A and Ypt1 is ~70%, and Rab1A can replace the essential function of Ypt1 in cell viability (Haubruck et al, 1989). Therefore, it was expected that Rab1A and Rab1B would also overlap in function. Indeed, whereas single KOs of human Rab1A and Rab1B in tissue culture cells do not cause lethality, a double KO of both is lethal, indicating an essential functional overlap between them (Blomen et al, 2015; Homma et al, 2021). Because secretion is required for cell viability, it was expected that at least one Rab1 paralog is needed for secretion. Indeed, knockdown (KD), KO, or dominant interfering Rab1A/B mutations were shown to affect Golgi stability and secretion (Zhang et al, 2009; Halberg et al, 2016; Liu et al, 2016). As for the role of Rab1A and Rab1B in autophagy, studies using gene KD or dominant interfering mutations have yielded conflicting results (Zoppino et al, 2010; Huang et al, 2011; Kakuta et al, 2017; Song et al, 2018).

Defects in the secretory and autophagy pathways and in the function of Rab GTPases result in a wide range of diseases (Kiral et al, 2018; Guadagno & Progida, 2019; Jin et al, 2021). Rab1A/B was specifically implicated in cancer (Thomas et al, 2014; Xu et al, 2015; Halberg et al, 2016) and neurodegenerative disease (Winslow et al, 2010; Coune et al, 2011). Therefore, the question whether Rab1A and Rab1B play a role in secretion and autophagy like Ypt1 is important

Department of Biochemistry and Molecular Genetics, University of Illinois at Chicago, Chicago, IL, USA

Correspondence: nava@uic.edu

for understanding the mechanisms of their pathogenicity and for designing therapeutic strategies for such diseases.

Here, we determined the controversial roles of Rab1A and Rab1B in autophagy using CRISPR-constructed KOs and exogenously expressed proteins. We show that Rab1A, but not Rab1B, plays a role early in the stress-induced autophagy pathway and localizes to APs. Switching the HVD of Rab1B with that of Rab1A enables Rab1B to localize to APs and function in autophagy. These results define a novel role for the HVD of Rab GTPases in granting a single Rab dual functionality in two very different processes, secretion, and autophagy.

## Results

We assembled a set of KOs of Rab1A, Rab1B, and Atg12 (as a control for the autophagy experiments) genes in two different human cell lines, HEK293T and HAP1. HEK293T (HEK) is derived from embryonic kidney cells (Kavsan et al, 2011) and HAP1 is a near-haploid cell line derived from cancerous myeloid cells (Carette et al, 2009). KOs were generated by us or by others using CRISPR technology (see the Materials and Methods section) and were validated by us using sequencing and immunoblot analysis with specific antibodies. Specifically, we confirmed that in both cell lines, the Rab1A protein, but not Rab1B, is missing in Rab1AKO; Rab1B protein, but not Rab1A, is missing in Rab1BKO; and Atg12 protein is missing in Atg12KO cells (Fig S1). Rab1AKO and Rab1BKO cell lines grow in rates similar to those of WT cells.

To ensure that the relevant phenotypes are caused by the KO, Rab1AKO cells were complemented with exogenously expressed N-terminally tagged proteins using two different tags and transfection modes. First, cells were transfected with constructs for the expression of Myc-Rab1A or HA-Rab1B to generate stable transformants. As an alternative for complementation, we used transient transfection with GFP-tagged Rab1A and Rab1B at their N terminus. Expression of tagged Rab1A and Rab1B was verified using immunoblot analysis and specific antibodies against the tag (Fig S2A, B, and F) and the Rab1 (Fig S2C, D, H, and I). Quantification of immunoblots using anti-Rab1A/B antibodies indicated that the stably transfected Myc-Rab1A and HA-Rab1B were expressed to a similar level and ~ninefold compared with the endogenous Rab1A/B (Fig S2E). The GFP-Rab1A and GFP-Rab1B were expressed to a similar level and ~13-fold compared with the endogenous Rab1A/B. Fluorescence microscopy showed ~60% transfection of the GFP-Rab1A or GFP-Rab1B in WT and KO cell lines (see an example in Fig S2G). The KO cell lines and their complemented versions were used in the experiments described below.

### Effect of Rab1AKO and Rab1BKO on secretion

A complete defect in secretion is predicted to result in lethality (Feyder et al, 2015). Because the single Rab1AKO and Rab1BKO are viable, it is expected that secretion defects would be partial, similar to those seen in KD experiments (Zhang et al, 2009; Halberg et al, 2016; Liu et al, 2016). We wished to confirm this point and also use complementation of such partial defects to validate the functionality of exogenously expressed tagged Rab1A and Rab1B. These

experiments were performed in HEK cells using two different assays: Golgi fragmentation and secretion. Golgi morphology was determined by immunofluorescence microscopy following the cis Golgi marker GM130. The Golgi in WT cells is compact and only <20% of the cells show a less-compact Golgi. The drug brefeldin A (BFA), which causes complete fragmentation of the Golgi (Alvarez & Sztul, 1999), was used here for all the cell lines to visualize complete Golgi fragmentation. Most Rab1AKO and Rab1BKO cells exhibited fragmented Golgi (~60% and 80%, respectively), although not as severely fragmented as when BFA was added (Fig 1A and B). The luciferase secretion assay (Kumar et al, 2016) showed a defect only in Rab1AKO, but not Rab1BKO, cells when compared with WT cells. This secretion defect is partial when compared with the complete secretion block caused by BFA (Fig S3A). Thus, both Rab1AKO and Rab1BKO cells exhibit partial defects in the secretory pathway.

The partial Golgi fragmentation and secretion phenotypes of the Rab1AKO cells were used to test exogenously expressed and tagged Rab1A and Rab1B proteins. Both secretory phenotypes were complemented in Rab1AKO cells stably transfected with Myc-Rab1A or HA-Rab1B (Figs 1C and D and S3B). The partial Golgi fragmentation phenotype of Rab1AKO cells was also complemented by transiently expressed GFP-Rab1A and GFP-Rab1B (Fig 1E and F). In addition, Ypt1 localizes to the yeast Golgi (Kim et al, 2016) and Rab1A and Rab1B localize to the Golgi in HeLa cells (Monetta et al, 2007; Dong et al, 2012). As expected for functional Rab1 proteins, both GFP-Rab1A and GFP-Rab1B localize to the Golgi in WT HEK cells as shown by their colocalization with the Golgi marker GM130 using fluorescence microscopy (Fig S3C and D).

Together, these results confirm that both Rab1A and Rab1B GTPases play a role in the secretory pathway because the KO of each shows a partial defect, and the Rab1AKO defect can be complemented by either Rab1A or Rab1B. These results also verify that the N-terminally tagged Rab1A and Rab1B, with a small (Myc and HA, ~1 kD) or large tag (GFP, ~28 kD), are functional.

### Effect of Rab1AKO, and not Rab1BKO, on stress-induced autophagy

While KD of Rab1A was reported to affect autophagy, reports about Rab1B KD were conflicting (Winslow et al, 2010; Zoppino et al, 2010; Huang et al, 2011; Kakuta et al, 2017; Song et al, 2018). Specifically, even in the same cell line, HeLa, depletion of Rab1B showed opposite effects on autophagy (Winslow et al, 2010; Zoppino et al, 2010; Kakuta et al, 2017). We wanted to determine the effect of complete depletion of each of these two very similar proteins (92% identity at the amino acid sequence) on stress-induced autophagy. Autophagy was induced by a serum and amino-acid starvation in Earle's Balanced Salt Solution (EBSS) medium (Martinet et al, 2006) or inhibition of mTORC1 by rapamycin (Sarkar et al, 2009). Two different approaches, each employing two endogenously expressed markers, were used to determine defects in autophagy: fluorescence microscopy using two AP markers, LC3 and p62, and a lysosomal marker LAMP, and immunoblot analysis of two proteins, LC3 and p62 (Jiang & Mizushima, 2015; Klionsky et al, 2021). In addition to being an LC3 adaptor, p62 plays other roles beyond autophagy and localizes to cellular components other than APs, for example, protein aggregates (Moscat & Diaz-Meco, 2009). Therefore,

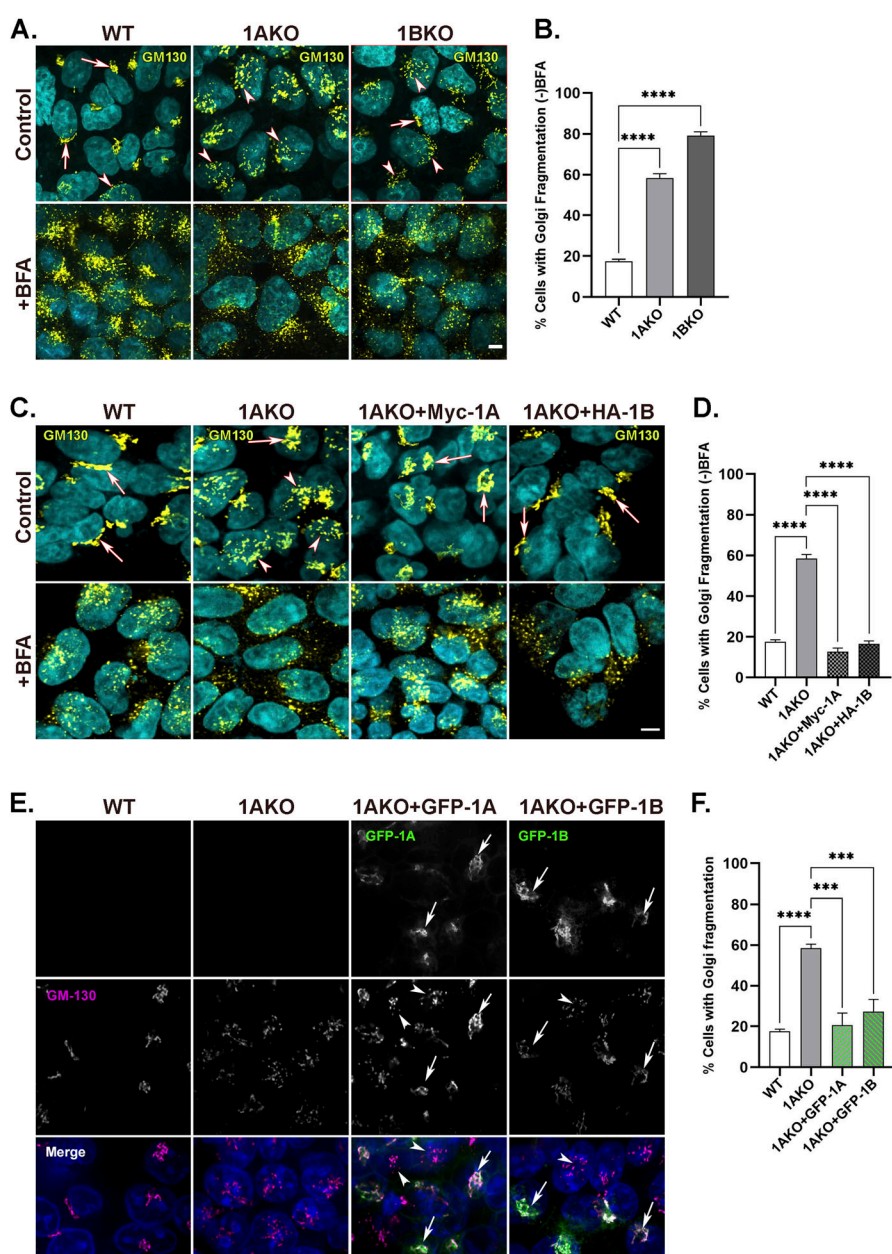

**Figure 1. Rab1A and Rab1B play a role in Golgi morphology.**
**(A, B)** Knockout of Rab1A or Rab1B results in Golgi fragmentation. **(A)** WT, Rab1AKO, and Rab1BKO HEK293T cells (from left to right), without (top) or with BFA (10 µg/ml for 1 h, bottom, control for complete Golgi fragmentation), were fixed with PFA and visualized by immunofluorescence microscopy using anti-GM130 antibody (yellow, Golgi) and DAPI (cyan, nucleus); arrows pointing at cells with intact Golgi, arrowheads pointing at cells with fragmented Golgi. **(A, B)** Bar graph showing percent of cells with Golgi fragmentation (from panel (A)) as mean ± SD of four independent experiments, (****$P < 0.0001$). **(C, D)** The Golgi fragmentation phenotype in Rab1AKO cells can be rescued by Myc-Rab1A and HA-Rab1B. **(C)** GM130 immunostaining of WT, Rab1AKO, and Rab1AKO stably transfected with Myc-Rab1A or HA-Rab1B (left to right). **(A)** Cells were not treated (top) or treated with BFA (bottom) and the Golgi apparatus was visualized as in panel (A). Arrows point to cells with intact Golgi, and arrow heads indicate cells with fragmented Golgi. **(C, D)** Bar graph showing percent of cells with Golgi fragmentation (from panel (C)); data represent the means ± SD from three independent experiments. **(E, F)** The Golgi fragmentation phenotype in Rab1AKO cells can be rescued by GFP-Rab1A and GFP-Rab1B. **(A, E)** Control WT, Rab1AKO, and Rab1AKO transfected with GFP-Rab1A and GFP-Rab1B (from left to right) were fixed with methanol and visualized by immunofluorescence microscopy as in panel (A). From top to bottom: GFP (green), GM130 (magenta), and merge (white). Arrows indicate GFP-Rab1A or GFP-Rab1B colocalization with GM130, mostly intact Golgi; arrowheads indicate fragmented Golgi in un-transfected Rab1AKO. **(E, F)** Percent cells with fragmented Golgi from panel (E) (for Rab1AKO + GFP-1A or GFP-1B, only transfected cells were considered; green bars) was determined by evaluating a minimum of 50 cells, two independent experiments (***$P < 0.001$, ****$P < 0.0001$).

we used p62 in microscopy only in combination with LC3, and in immunoblot analysis as previously recommended (Bjorkoy et al, 2009). The effect of Rab1AKO and Rab1BKO on autophagy was determined in two different cell lines, HEK and HAP1. The Atg12KO in HEK and HAP1 cell lines were used as a negative control for a complete autophagy defect due to a block in LC3 lipidation, which prevents attachment of LC3 to the AP membrane (Tanida et al, 2004).

### Fluorescence microscopy assays
The microscopy assays we used determine two different events in stress-induced autophagy. Under stress, the number of LC3 puncta increases and its colocalization with the LC3 adaptor p62, representing

AP formation, also increases (Bjorkoy et al, 2009). Colocalization of LC3 or p62 with the lysosomal marker LAMP shows delivery of autophagy cargo to the lysosome, or autophagy flux (Klionsky et al, 2021). To ensure that the phenotype is caused by a defect en route to the lysosome, we used a drug that blocks lysosomal proteases, BafA1 (Drose & Altendorf, 1997). As expected, unstressed (untreated) WT cells show low numbers of LC3 puncta, colocalized LC3/p62 puncta, and LC3/LAMP puncta, defining the background levels of these phenotypes during normal cell growth. Untreated Rab1AKO cells behaved like WT cells. Atg12KO cells do not have any such puncta (HEK, Fig S4; HAP1, Fig S5A–C) because in these mutant cells, LC3 is not lipidated and therefore cannot attach to membranes and

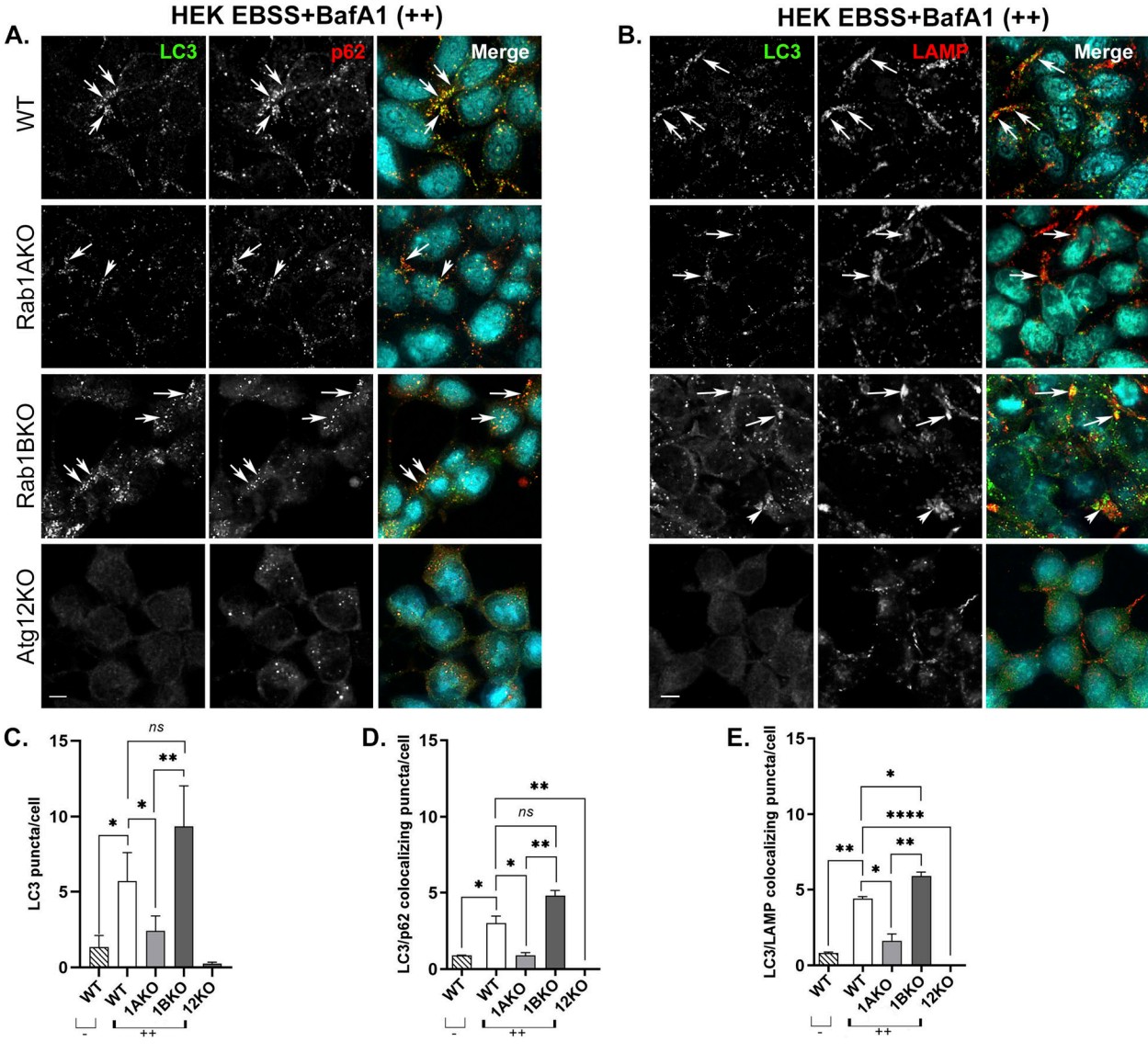

**Figure 2. Rab1AKO, but not Rab1BKO, cells are defective in EBSS-induced autophagy by microscopy (HEK293T cells).**
**(A, B)** HEK293T cells from top to bottom: WT, Rab1AKO, Rab1BKO, and Atg12KO (as a negative control) were incubated in complete medium; after 24 h, the cells were starved in EBSS medium for 3 h (stress), and 100 nM BafA1 was added during the last 1.5 h of treatment (to block degradation in the lysosome). **(A, B)** Cells were fixed with methanol and visualized by immunofluorescence microscopy: (A) AP proteins: from left to right: LC3 (green), p62 (red), nuclei stained with DAPI (cyan), and merge; (B) Autophagy flux: from left to right: LC3 (green), LAMP (lysosomal membrane, red), nuclei stained with DAPI (cyan), and merge. **(A, B)** Arrows point to LC3/p62 (A) and LC3/LAMP (B) colocalizing puncta. **(A, B, C, D, E)** Quantification of results shown in panels (A, B) and Fig S4 (control of untreated cells). **(C, D, E)** Bar graphs showing LC3 puncta/cell (C), LC3/p62 colocalizing puncta/cell (D), and LC3/LAMP colocalizing puncta/cell (E). The three autophagy phenotypes determined are as follows: upon starvation and with BafA1 (++), the number of LC3 puncta (APs), LC3 puncta colocalizing with p62 (APs), and LC3 puncta colocalizing with LAMP (autophagy cargo in the lysosome) increase in WT when compared with untreated cells (−). Rab1BKO, but not Atg12KO, cells show a similar increase, and the increase is significantly lower in Rab1AKO cells. >80 cells were quantified, the number of puncta presents the mean ± SD, (*$P < 0.05$, **$P < 0.01$, ****$P < 0.0001$, *ns* – not statistically significant), scale bar, 10 $\mu$m. Results in this figure represent three independent experiments.

form puncta. Interestingly, untreated Rab1BKO HEK cells show significantly more LC3 puncta and colocalizing LC3/LAMP when compared with WT cells (Fig S4A–C and E). This last observation is discussed below.

In WT HEK, but not in Atg12KO cells, under stress with EBSS or rapamycin, the number of LC3 puncta and LC3 colocalization with p62 or with LAMP significantly increases (Figs 2 and S6A–C). Under these stresses, Rab1AKO, but not Rab1BKO HEK, cells exhibits three significant autophagy defects: low number of LC3 puncta, less LC3/p62 colocalization, and less LC3/LAMP colocalization (Figs 2 and S6A–C). Similarly, in HAP1 cells stressed wither with EBSS or rapamycin, Atg12KO conferred complete blocks of all autophagy phenotypes, and Rab1AKO elicited significant defects of LC3 puncta and LC3 colocalization with LAMP when compared with WT cells. In contrast, Rab1BKO in HAP1 cells under either stress exhibited no autophagy defects in the number of LC3 puncta or LC3/LAMP colocalization (Figs S5D–F and S6D–F).

## A. HEK EBSS+BafA1

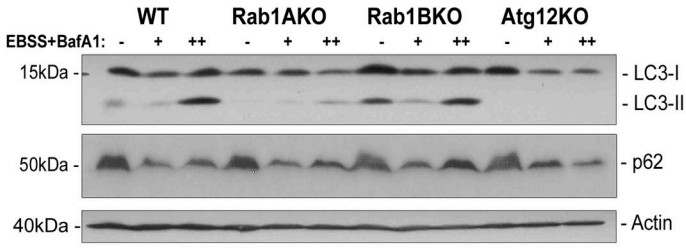

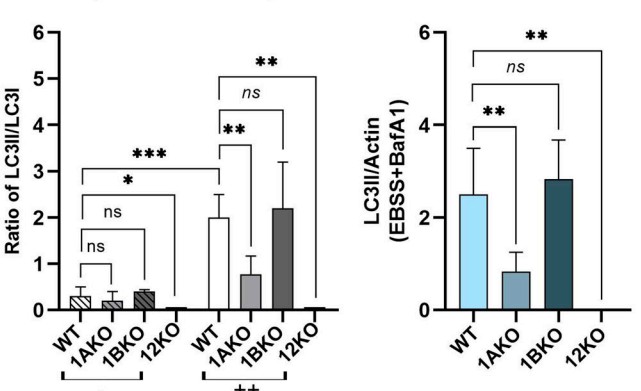

## B. LC3 (EBSS+BafA1)

## C. LC3II

## D. p62 (EBSS)

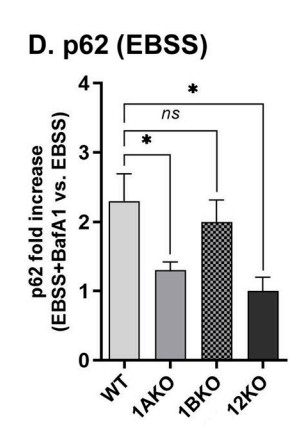

## E. HAP1 EBSS/Rapamycin+BafA1

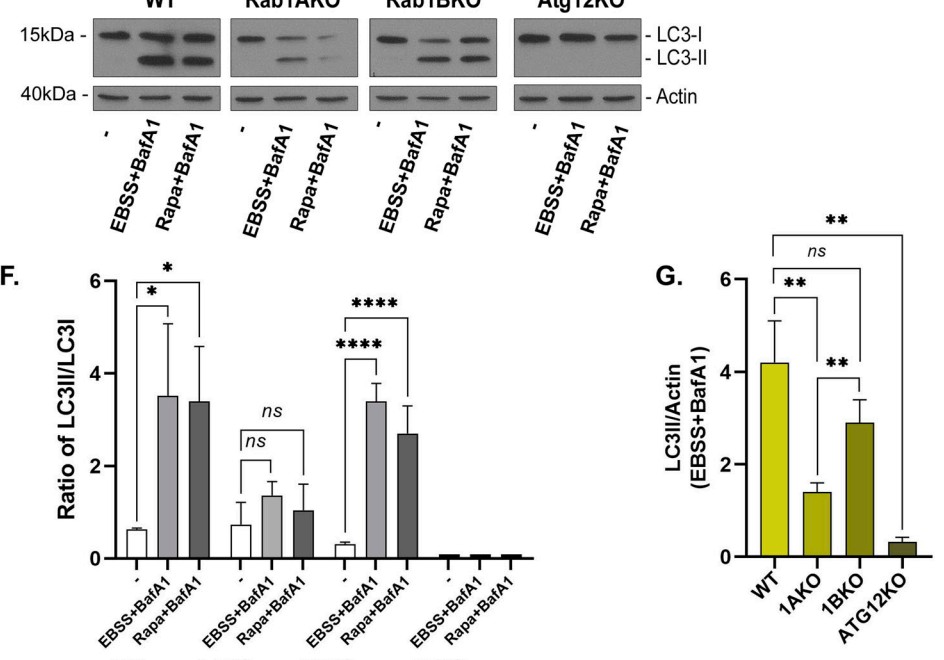

## F.

## G.

**Figure 3. Rab1AKO, but not Rab1BKO cells, are defective in EBSS-induced autophagy using immunoblot analysis (HEK293T and HAP1 cells).**
**(A, B, C, D)** HEK293: WT, Rab1AKO, Rab1BKO, and Atg12KO cells were incubated in full medium (−), EBSS for 3 h (+), or EBSS for 3 h with the addition of BafA1 (100 nM) for the last 1.5 h (++). Whole-cell lysates were collected and subjected to immunoblot analysis using anti-LC3 and anti-p62 antibodies (anti-actin for a loading control). **(A)** Immunoblots are shown from left to right: molecular weight markers (kD); three lanes for each strain (−) (+) (++): WT, Rab1AKO, Rab1BKO, and Atg12KO; and detected protein. Top to bottom: cell line, growth conditions, LC3 blot, p62 blot, and actin blot (loading control). **(A, B)** Bar graphs of quantified results from panel (A) showing the ratio (within lanes) of LC3II (lipidated) to LC3I (unlipidated) in untreated cells (−, left) and in cells treated with EBSS+BafA1 (++, right). **(A, C)** Bar graphs of quantified results from panel (A) showing the level of LC3II (lipidated) corrected to the loading control (actin), in cells treated with EBSS+BafA1. **(A, D)** Bar graph of quantified results from panel (A) showing the fold increase of the p62 level in cells treated with EBSS and BafA1 (++) compared with cells treated in EBSS alone (+) (corrected to the loading control, actin). **(A, E, F, G)** HAP1: WT, Rab1AKO, Rab1BKO, and Atg12KO cells were treated with EBSS or rapamycin (200 nM for 3 h) in the presence of BafA1 during the last 1.5 h of treatment. Immunoblot analysis using anti-LC3 was done as described in panel (A). **(E)** Immunoblots using anti-LC3 antibodies (anti-actin for a loading control). Shown from left to right: molecular weight markers (kD), WT, Rab1AKO, Rab1BKO, Atg12KO: three lanes for each cell line: no treatment (−), EBSS+BafA1, rapa+BafA1; detected protein. **(E, F)** Bar graph of quantified results from panel (E) showing the ratio of protein levels LC3II/lC3I. **(E, G)** Bar graphs of quantified results from panel (E) showing the level of LC3II (lipidated) corrected to the loading control (actin), in cells treated with EBSS+BafA1. Upon EBSS+BafA1 (HEK293 and HAP1 cells) or rapamycin (HAP1 cells) treatments, LC3II (HEK293 and HAP1 cells) and p62 (HEK293 cells) levels are increased in WT and Rab1BKO cells, but not in Atg12KO or Rab1AKO cells. Values in the bar graphs are presented as mean ± SD (*$P < 0.05$, **$P < 0.01$, ***$P < 0.001$, ****$P < 0.0001$, ns − not statistically significant). Results in this figure represent three independent experiments.

### Immunoblot assay

Here we used two markers, LC3 and p62. Under stress, LC3 is lipidated by Atg5, 7, and 12 to attach to the AP membrane (Tanida et al, 2004). The unlipidated, LC3I, and lipidated, LC3II, forms of LC3 can be detected by immunoblot analysis using anti-LC3 antibodies (Jiang & Mizushima, 2015). During autophagy, if lysosomal proteases are blocked, in WT cells, but not in cells defective in LC3 lipidation (e.g., Atg12KO), LC3-II accumulates. The increase in LC3II level was

determined by two ways: the ratio of LC3II to LC3I in the same lane, and comparing LC3II levels between lanes while adjusting to the loading control (Sharifi et al, 2015). We also looked at the effect of stress on another protein, p62, which is expressed in untreated cells and degraded under stress. The degradation of p62 under stress through autophagy can be rescued by addition of BafA1, which blocks lysosomal proteases. In this assay, the level of p62 in cell stressed by EBSS is compared with its level in cells stressed with EBSS in the presence of BafA1. In WT cells, but not in cells defective in autophagy, this ratio increases (Bjorkoy et al, 2009; Fernandez, 2018).

WT cells accumulate LC3II under stress, especially when lysosomal proteases are blocked by a drug (BafA1). Such a significant increase can be seen in HEK cells stressed with EBSS and HAP1 cells treated with EBSS or rapamycin. As expected, Atg12KO cells, HEK or HAP1, did not show any LC3II band. Rab1AKO cells are not defective in LC3 lipidation and therefore show some LC3II, but this LC3 form is significantly lower than in WT cells. In Rab1BKO cells, HAP1 and HEK do not exhibit any defect in accumulation of LC3II under stress (HEK+EBSS, Fig 3A–C; HAP1+EBSS or rapamycin, Fig 3E–G). Following p62 degradation in HEK cells also reveals a significant autophagy block in Rab1AKO and Atg12KO cells, but not in Rab1BKO, when compared with WT HEK cells (Fig 3A and D).

Together, the microscopy and immunoblot assays establish that Rab1AKO, but not Rab1BKO, cells are defective in stress-induced autophagy. The microscopy assay that follows LC3/p62 colocalization shows that this Rab1AKO defect is in an early step of autophagy, the assembly of APs. Interestingly, Rab1BKO HEK cells, not only do not exhibit an autophagy defect in the microscopy assay, but the number of LC3 puncta is significantly higher in untreated Rab1BKO than in WT cells (Fig S4C and E; see the Discussion section).

## Complementation of the Rab1AKO autophagy phenotype by Rab1A and not Rab1B

We performed complementation analysis of the autophagy phenotypes of Rab1AKO cells for two reasons: first, to confirm that the Rab1AKO phenotype was caused by Rab1A depletion; second, to exclude the possibility that Rab1B can play a minor role in autophagy even though it is not seen in its KO. We used two independent ways to express exogenous Rab1A and Rab1B: Stable transfection of Myc-Rab1A and HA-Rab1B and transient transfection with GFP-tagged Rab1A and Rab1B. All these constructs were shown to be expressed (Fig S2), their GFP-tagged versions localized to the Golgi in WT cells (Fig S3C and D), and they are functional because they can complement the secretory defects of Rab1AKO cells (Figs 1C–F and S3B). Complementation was determined using fluorescence microscopy for the stably and transiently transfected cells and immunoblot analyses for the stably transfected cells.

Using Rab1AKO HEK cells stably transfected with Myc-Rab1A or HA-Rab1B in the microscopy assays, Myc-Rab1A, but not HA-Rab1B, can fully complement the EBSS-induced Rab1AKO phenotypes: the number of LC3 puncta, LC3/p62 colocalizing puncta, and LC3/LAMP colocalizing puncta (Fig 4). Rab1A can also complement the Rab1AKO LC3/LAMP colocalization phenotype in HAP1 cells (Fig S7A and B). Using the immunoblot analysis, Myc-

Rab1A, and not HA-Rab1B, can fully complement the EBSS-induced Rab1AKO phenotypes in HEK cells: LC3II and p62 accumulation upon addition of BafA1 (Fig 5). In HAP1 cells, Myc-Rab1A can also fully complement the LC3II accumulation block of Rab1AKO cells (Fig S7C and D).

Using Rab1AKO HEK transiently transfected with GFP-Rab1A or GFP-Rab1B in the microscopy assays, GFP-Rab1A, and not GFP-Rab1B, can fully complement the EBSS-induced Rab1AKO phenotypes: the number LC3/p62 colocalizing puncta and LC3/LAMP colocalizing puncta (Fig 6). We constructed a modified version of GFP-Rab1B for the following reason: GFP-Rab1A and GFP-Rab1B are mouse proteins (Ishida et al, 2012). Whereas the Rab1A mouse (m) and human (h) proteins are identical, Rab1B has one amino-acid difference in position 197 (S197 and G197, respectively). We replaced the S197 in GFP-mRab1B with 197G to rule out the possibility that the GFP-mRab1B we use is not functional in human cells because of this difference. GFP-Rab1B-197G was used for complementation of the Rab1AKO autophagy phenotype. Like GFP-mRab1B, GFP-Rab1B-197G cannot complement the autophagy phenotypes: the number of LC3 puncta and colocalization of LC3/p62 (Fig 7). Therefore, the difference between mouse and human Rab1B in the 197 residue is irrelevant to the inability of GFP-Rab1B to complement the Rab1AKO autophagy phenotype.

These results show that exogenously expressed Rab1A, but not Rab1B, tagged with two different tags, can complement the autophagy phenotypes of Rab1AKO cells. Together with the finding that Rab1AKO, and not Rab1BKO, cells exhibit autophagy phenotypes, and because exogenously expressed Rab1B was shown to be functional in the secretion assays, we can conclude that Rab1A, and not Rab1B, functions in autophagy.

## Rab1A/B HVD swap

To get to the mechanism of how Rab1A, and not Rab1B, functions in autophagy, we looked at its amino acid sequence. Whereas Rab1A (205 amino acids) and Rab1B (201 amino acids) share 92% identity at the amino acid sequence, half of the nonidentical amino acids between the two proteins (8/16) are in the last 24 amino acids at the HVD at the Rab1B C terminus (residues 178–201) and the 25 amino acids of Rab1A (181–205) (Fig 7A). This HVD domain is the most variable protein domain among Rab GTPases and is considered to be important for the localization of different Rabs to specific intracellular compartments (Chavrier et al, 1991; Li et al, 2014). To determine if this domain is important for Rab1A function in autophagy, the HVD of GFP-Rab1B was replaced with that of Rab1A to generate the chimera, GFP-Rab1B-HVD-1A (Rab1B 1–177 plus Rab1A 181–205, a total of 202 amino acids). Immunoblot analysis using anti-GFP antibodies showed that the GFP-tagged Rab1A, Rab1B, the Rab1B-chimera (and the Rab1B-197G) were expressed to a similar level (Fig 7B and C). The chimera was tested for its ability to complement the autophagy phenotype of Rab1AKO (in HEK cell) using fluorescence microscopy.

Whereas GFP-Rab1B and GFP-Rab1B-197G do not complement the autophagy phenotypes of Rab1AKO cells, the number

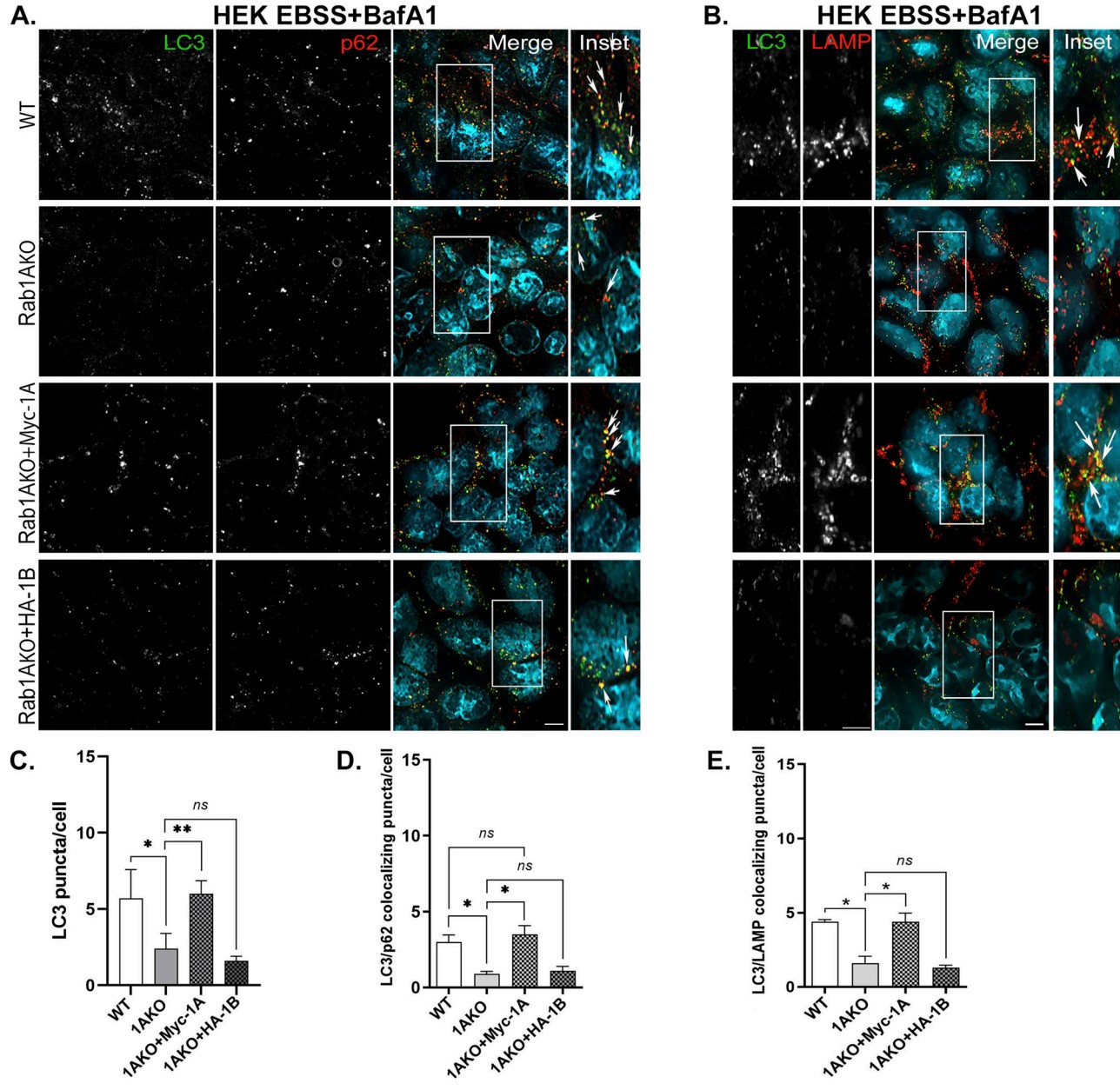

**Figure 4. Rescue of autophagy phenotypes of Rab1AKO by Myc-Rab1A, but not by HA-Rab1B: Microscopy.**
**(A, B)** Rab1AKO HEK293T cells were stably transfected with Myc-Rab1A or HA-Rab1B (tagged at their N terminus). Cells were treated for 3 h with EBSS in the presence of BafA1. Cells were then fixed with methanol followed by immunostaining with LC3, p62, and LAMP antibodies. Shown from top to bottom: WT, Rab1AKO, Rab1AKO+Myc-Rab1A, and Rab1AKO+HA-Rab1B. **(A)** AP proteins from left to right: LC3, p62, merge, and inset (enlarged view from inset in merge); arrows point to LC3/p62 colocalization (autophagosomes). **(B)** Autophagy flux from left to right: LC3, LAMP, merge, and inset (enlarged view from inset in merge); arrows indicate the LC3/LAMP colocalization (autophagosomes inside lysosomes). **(A, B, C, D, E)** Quantification of results from panels (A, B): the LC3 puncta average number (C), LC3/p62 (D) and LC3/LAMP colocalizing puncta/cell (E). **(C, D, E)** Myc-Rab1A, but not HA-Rab1B, complements the autophagy phenotypes of Rab1AKO: number of LC3 puncta (C), number of LC3/p62 colocalizing puncta (D), and the number of LC3/LAMP colocalizing puncta (E). Puncta numbers are presented as mean ± SD from three independent experiments with >60 cells quantified for each cell type (*P < 0.05, **P < 0.01, ns – not statistically significant).

of LC3 puncta and LC3/p62 colocalizing puncta, the GFP-Rab1B-HVD-1A chimera (1BKO+Chi) complements these phenotypes to the WT levels like GFP-Rab1A (Fig 7D–F). Thus, replacing the 8/16 nonidentical residues between Rab1A and Rab1B at the HVD is enough to render a Rab1B protein functional in autophagy. We conclude that the HVD of Rab1A is important for the specific role it plays in stress-induced autophagy.

## HVD-dependent Rab1A localization to AP

The HVD of Rabs is important for their proper intracellular localization (Li et al, 2014). We tested whether GFP-Rab1A and the GFP-Rab1B-HVD-1A chimera (1BKO + Chi) colocalize with the AP marker LC3 under stress. In Rab1AKO cells (HEK) stressed with EBSS, both GFP-Rab1A and the GFP-Rab1B-HVD-1A chimera

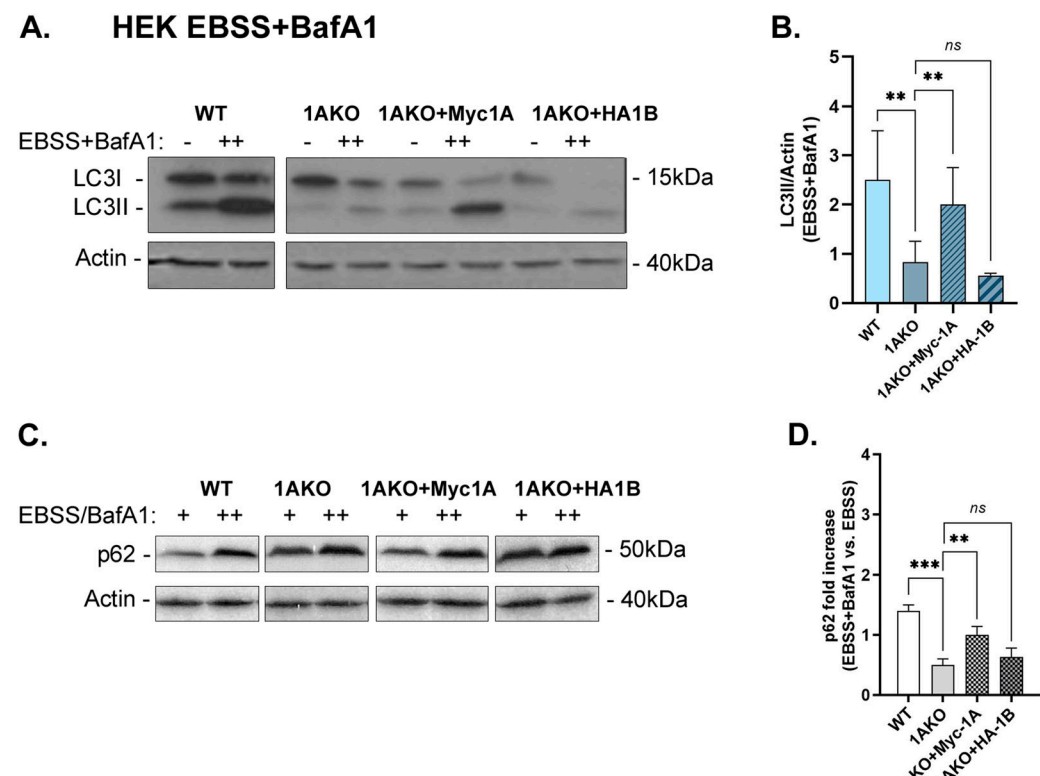

**Figure 5. Rescue of autophagy phenotypes of Rab1AKO by Myc-Rab1A, but not by HA-Rab1B: Immunoblot analysis.**
Rab1AKO HEK293T cells were transfected with Myc-Rab1A or HA-Rab1B (tagged at their N terminus). Cells were grown in complete medium (−) or treated for 3 h with EBSS in the presence of BafA1 (++). **(A, C)** Whole-cell lysates were collected and subjected to immunoblot analysis using anti-LC3 (A) and anti-p62 (C) antibodies (anti-actin for loading control). **(A, C)** Immunoblots shown from left to right: detected protein, blots: WT, Rab1AKO, Rab1AKO+Myc-Rab1A, and Rab1AKO+HA-Rab1B; doublet for each cell line: (−) and (++); molecular weight markers (kD). **(A, C)** Top to bottom: Cell line, growth conditions, LC3 (A) or p62 (C) blot, actin blot. **(A, B)** Bar graphs of quantified results from panel (A) showing the level of LC3II (lipidated) corrected to the loading control (actin), in cells treated with EBSS+BafA1. Myc-Rab1A, but not HA-Rab1B, rescues the low LC3II level in Rab1AKO cells. **(C, D)** Bar graph showing quantification from panel (C) showing the fold increase of p62 level (corrected to the loading control, actin) in cells treated with EBSS+BafA1 (++) compared with treatment with EBSS (+). Myc-Rab1A, but not HA-Rab1B, rescues the low p62-level phenotype of the Rab1AKO. **(B, D)** Bar graphs showing the quantification as mean ± SD of three experiments for LC3II/actin (B) and two experiments for p62 (D), (**$P < 0.01$, ***$P < 0.001$, ns − not statistically significant).

colocalized with LC3 in ~40–50% of the cells. In contrast, GFP-Rab1B (and the GFP-Rab1B-197G) showed significantly less colocalization with LC3 (<20% of the cells) (Fig 8A and B and Video 1 and Video 2). These results suggest that the mechanism for the specific role of Rab1A in autophagy is determined by an HVD-dependent localization to APs.

# Discussion

The three important conclusions from the results we present here are as follows: first, although Rab1A and Rab1B, which share 92% amino-acid sequence identity, both contribute to the essential process of secretion during normal growth, only Rab1A plays a role in stress-induced autophagy; second, under stress, like Ypt1 in yeast, depletion of Rab1A results in an early autophagy block, AP formation (Fig 8C); third, the mechanism of the Rab1A versus Rab1B specificity is HVD-dependent regulation of autophagy and localization to APs. This defines a novel role for the HVD of Rabs in granting dual functionality to a single Rab, Rab1A, in two different trafficking pathways, secretion and autophagy.

## Rab1A-specific role in stress-induced autophagy

The finding that only Rab1A, and not Rab1B, functions in autophagy was surprising, considering the high similarity between these two Rabs (92%), especially since the yeast Ypt1, which shares lower similarity with these Rab1A/B (~70%), functions in autophagy (Lipatova et al, 2013). However, the evidence supporting this conclusion is robust. (i) Only Rab1AKO, but not Rab1BKO, results in autophagy phenotypes. This was shown in two cell lines (HEK and HAP1) using two different stresses (EBSS and rapamycin), two different assays (fluorescence microscopy and immunoblot analysis), and multiple markers (LC3, p62, and LAMP). (ii) Only Rab1A, but not Rab1B, can complement the autophagy phenotypes of Rab1AKO cells. This was shown using stable single colonies expressing Myc-Rab1A (in two cell lines) or HA-Rab1B, and transient transfection with GFP-Rab1A and GFP-Rab1B (in HEK cells). The tagged Rab1B proteins are functional because they can complement the Golgi fragmentation phenotype of Rab1AKO cells. (iii) GFP-Rab1A, but not GFP-Rab1B, localizes to APs.

As noted in the Results section, using the microscopy assay, Rab1BKO HEK cells not only do not exhibit an autophagy defect

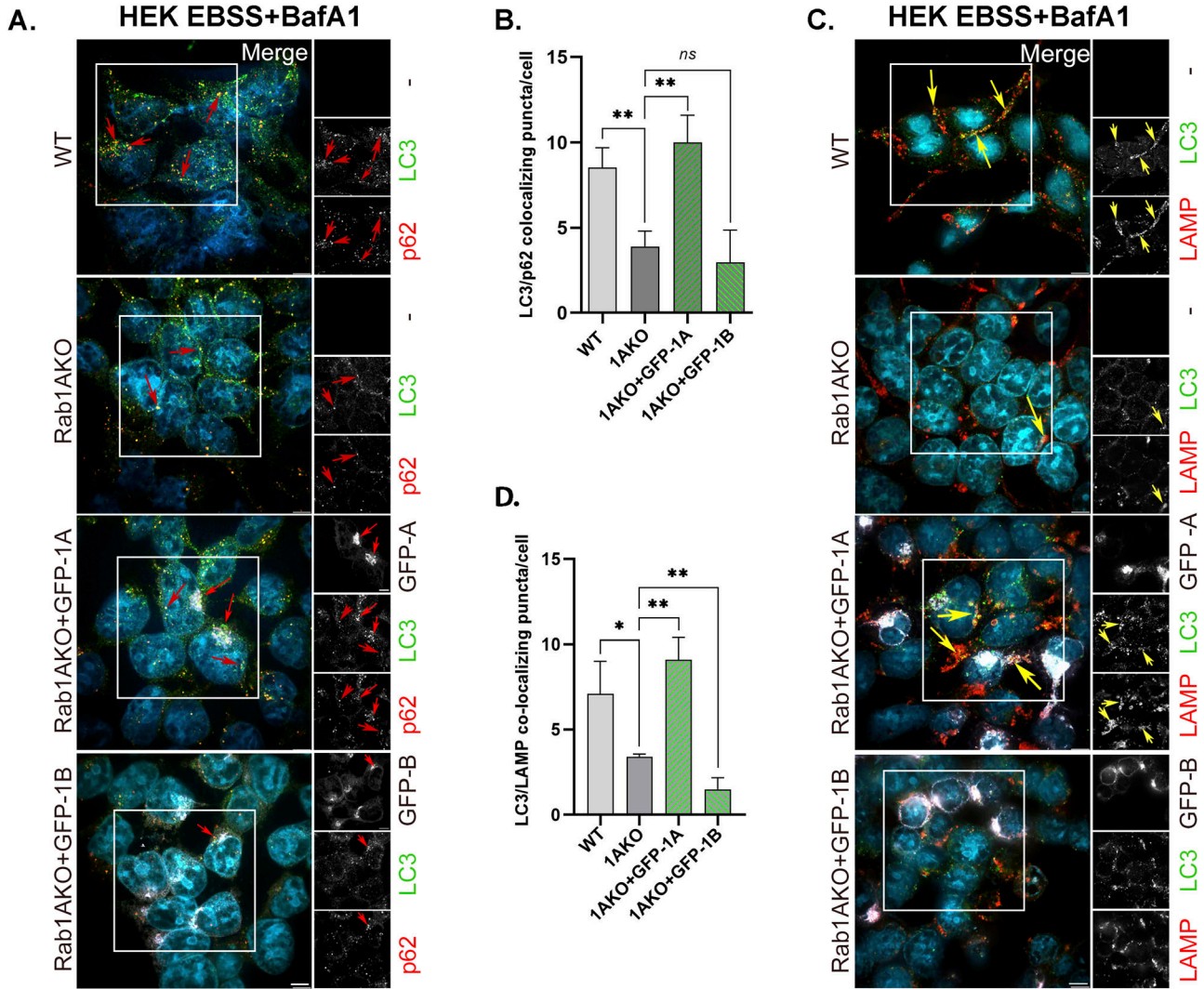

**Figure 6. Autophagy phenotype in Rab1AKO HEK293T cells can be rescued after transfection with GFP-Rab1A, but not with GFP-Rab1B.**
Rab1AKO HEK293T cells were transiently transfected with GFP-Rab1A or GFP-Rab1B–expressing plasmid. 24 h posttransfection (about 60% of the cells were transfected; see example in Fig S2G), the cells were incubated for 3 h in EBSS starvation medium in the presence of 100 nM BafA1 (during the last 90 min). **(A, C)** After treatment, the cells were fixed with methanol and immunostained for endogenous AP proteins, LC3, and p62 (A), or for autophagy flux, LC3, and LAMP (C). Shown from top to bottom: WT, Rab1AKO, Rab1AKO+GFP-Rab1A, Rab1AKO+GFP-Rab1B. **(A)** Shown from left to right: cell line, merge with inset, insets from three channels: GFP (top), LC3 (middle), and p62 (bottom) shown in black and white. Merge: artificial coloring in the merge: GFP, white; LC3, green; p62, red (DAPI, cyan). Red arrows point to colocalizing LC3/p62 puncta in un-transfected cells; in Rab1AKO+GFP-Rab1A/B, arrows point to colocalizing puncta only in transfected cells. **(A, B)** Bar graph showing the quantification of LC3/p62 colocalizing puncta per cell from panel (A): WT and Rab1AKO bars are from un-transfected cells, whereas green bars show the colocalization only in GFP-transfected Rab1AKO cells. The low LC3/p62 colocalization in Rab1AKO cells (when compared with WT cells) is rescued in cells transfected with GFP-Rab1A, but not GFP-Rab1B. **(C)** Shown from left to right: cell line, merge with inset, insets from three channels: GFP (top), LC3 (middle), and LAMP (bottom) shown in black and white. Merge: artificial coloring in the merge: GFP, white; LC3, green; LAMP red (DAPI, cyan). Yellow arrows point to LC3/LAMP colocalization in un-transfected cells; in Rab1AKO+GFP-Rab1A/B, arrows point to colocalizing puncta only in transfected cells. **(C, D)** Bar graph showing quantification of LC3-LAMP colocalizing puncta per cell from panel (C): WT and Rab1AKO are from un-transfected cells, whereas green bars show the colocalization only in GFP-transfected cells. The low LC3-LAMP colocalization in stressed Rab1AKO cells (when compared with WT cells) is rescued in cells transfected with GFP-Rab1A, but not GFP-Rab1B. Scale bar (white, in the merge), 10 μm. Bar graphs show data from three independent experiments, (*$P < 0.05$, **$P < 0.01$, ns - not statistically significant).

but in unstressed cells, the number of LC3 puncta is higher in Rab1BKO than in WT cells. One possible explanation to this phenomenon is that Golgi fragmentation, which occurs in both Rab1AKO and Rab1BKO cells, was shown to be associated with an increase in LC3 puncta (Gosavi et al, 2018). Notably, Rab1AKO cells, in spite of having a fragmented Golgi, exhibit an autophagy defect.

## Rab1A role in an early autophagy step

In addition to showing a defect in autophagy flux (LC3-LAMP colocalization), we can conclude that Rab1A plays a role in AP formation, an early step in the autophagy pathway. This conclusion is based on immunofluorescence microscopy results. Specifically, the number of LC3 puncta and the number LC3-p62 colocalizing

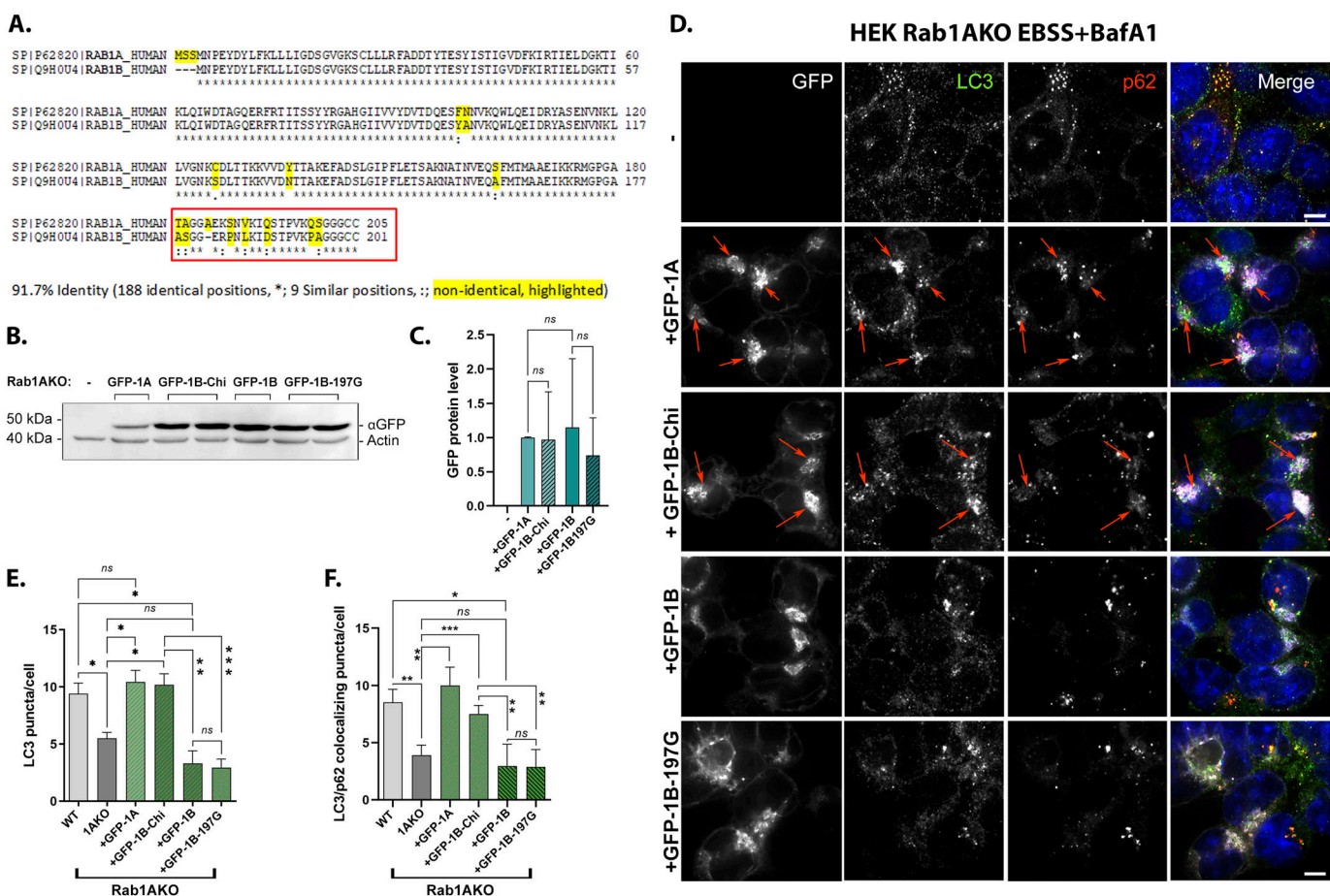

**Figure 7. The C-terminal hypervariable domain (HVD) of Rab1A is important for its function in autophagy and localization to APs.**
**(A)** Half (8/16) of the nonidentical residues of Rab1A and Rab1B are in their HVD. Sequence alignment of human Rab1A and Rab1B proteins by CLASTAL: Identical residues (*), similar (:), nonidentical (highlighted in yellow). **(B, C, D, E, F)** The red box shows the domain switched between Rab1A and Rab1B to generate the GFP-Rab1B-HVD-1A chimera (used in panels (B, C, D, E, F) and Fig 8A and B). **(B, C)** Similar levels of GFP-tagged Rab1A, Rab1B-chimera, Rab1B, and Rab1B-197G expressed in Rab1AKO-transfected cells. Rab1AKO (HEK293T) cells were transiently transfected with a plasmid for expression of GFP-Rab1A, GFP-Rab1B-chimera, GFP-Rab1B, or GFP-Rab1B-197G. Lysates from un-transfected or transfected cells were subjected to immunoblot analysis using anti-GFP (and anti-actin as a loading control). Shown from top to bottom, plasmid, and blot with anti-GFP and anti-actin antibodies. Shown from left to right: Mw markers, blot: un-transfected, GFP-1A, 2 lanes with GFP-1B-chimera, GFP-Rab1B, and two lanes with GFP-Rab1B-197G; and antibodies. The GFP-Rab1 bands with the expected size (~50 kD) are seen only in cells transfected these constructs. **(C)** Bar graph showing quantification of the different GFP-tagged Rab1 protein levels (corrected for the loading control, actin). The results in (C) are representative of three independent experiments. **(D, E, F)** The GFP-Rab1B-HVD-1A chimera (Chi) can rescue the autophagy phenotypes of Rab1AKO like GFP-Rab1A (whereas GFP-Rab1B-197G, like GFP-Rab1B, does not; see text). **(D)** Rab1AKO HEK cells were transiently transfected with: GFP-Rab1A, GFP-Rab1B-Chi, GFP-Rab1B, and GFP-Rab1B-197G. Cells were treated with EBSS+BafA1 (as in Fig 2A), fixed with methanol, and visualized by fluorescence microscopy with LC3 and p62 antibodies (and DAPI in blue to visualize nuclei, as described for Fig 2A). Shown from top to bottom: Rab1AKO (un-transfected), Rab1AKO+GFP-Rab1A, Rab1AKO+GFP-Rab1B-Chi, Rab1AKO+GFP-Rab1B, Rab1AKO+GFP-Rab1B-197G. Left to right: plasmid, GFP (white), LC3 (green), p62 (red), merge. Arrows point to colocalization of GFP with the autophagosome markers (LC3/p62). Scale bar, 10 μm. **(D, E)** Bar graph showing the number of LC3 puncta per cell from panel (D). **(D, F)** Bar graph quantification of the number of LC3/p62 colocalizing puncta per cell from panel (D). In (E, F): WT and Rab1AKO are from un-transfected cells (to show the AP formation phenotypes of Rab1AKO cells), whereas green bars show the number of puncta only in GFP-transfected Rab1AKO cells. Like GFP-Rab1A, GFP-Rab1B-chi complements the AP formation defects of Rab1AKO cells: the number of LC3 puncta and the number of LC3/p62 colocalizing puncta. GFP-Rab1B and GFP-Rab1B-G197G do not complement these phenotypes. **(E, F)** The data shown in panels (E, F) correspond to mean ± SD of four independent experiments (*P < 0.05, **P < 0.01, ***P < 0.001, ns – not statistically significant).

puncta decrease in Rab1AKO cells. These two criteria have been broadly used for determining defects in AP formation, the first step in stress-induced autophagy (Yoshii & Mizushima, 2017) and were previously used for determining a role for Atg9a in AP formation (Orsi et al, 2012).

The reason for the defect in accumulation of lipidated LC3, LC3II, seen in Rab1AKO cells using immunoblot analysis, is not immediately obvious. Whereas Atg12, Agt5, and Atg7, play a direct role in LC3 lipidation, which is conserved from yeast to human cells

(Ohsumi, 2014), Ypt1 and Rab1A are not expected to play such a role. In agreement, whereas Atg12KO cells show no accumulation of LC3II even under stress, Rab1AKO cells accumulate some LC3II in un-stressed and stressed cells. The Rab1AKO effect is similar to the effect of depletion of Atg9, a core autophagy protein that does not play a role in LC3 lipidation (Orsi et al, 2012). A possible explanation for the lower level of LC3II in Rab1AKO and Atg9KO cells when compared with that of WT is that LC3II attaches to the AP membrane through its lipid moiety and accumulates there. The expected lower

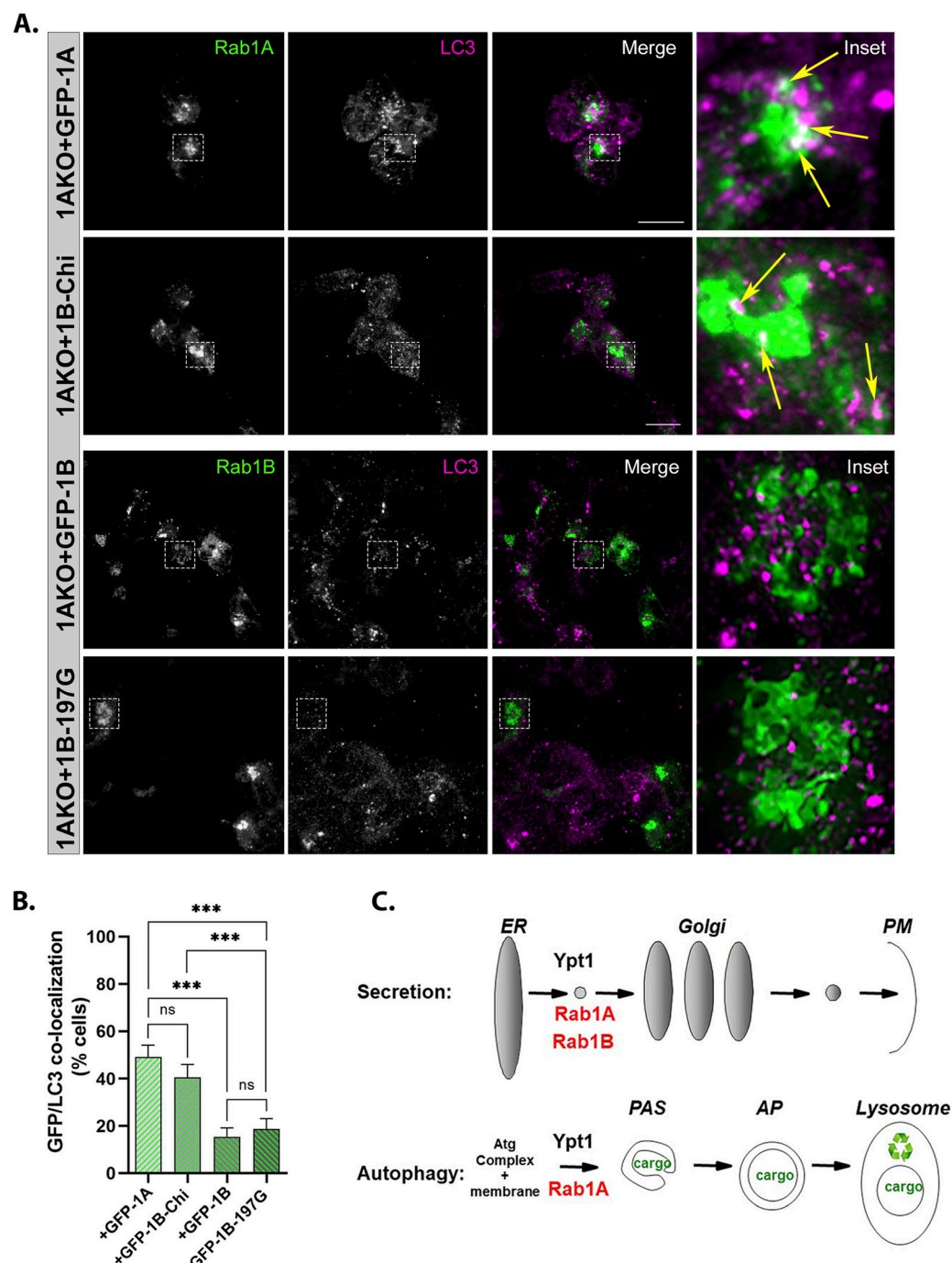

**Figure 8. The C-terminal HVD of Rab1A is important for its localization to APs.**
**(A)** GFP-Rab1A and the GFP-Rab1B-HVD-1A chimera, but not GFP-Rab1B or GFP-Rab1B-197G, colocalize with the AP marker LC3. Rab1AKO cells transiently transfected with (top to bottom): GFP-Rab1A, GFP-Rab1B-Chi, GFP-Rab1B, and GFP-Rab1B-197G, were treated with EBSS+BafA1 (as in Fig 2A), fixed with methanol and visualized using Z stack. Shown are 3D Z-stacks (left to right): GFP, LC3, merge, inset (enlarged view of inset in from merge). Arrows point to colocalization (white) of GFP (green) with the autophagosome marker LC3 (magenta). Scale bar, 20 μm. Also see Video 1 and Video 2. **(A, B)** Bar graph showing quantification of GFP-Rab1/LC3 colocalization (in cells from panel (A) and Fig 7D). Green bars show percent of transfected cells with GFP-LC3 colocalization. Significantly more colocalization is seen in cells transfected with GFP-Rab1A or Rab1B-Chi than in cells with GFP-Rab1B or Rab1B-197G. **(B)** The data shown in panel (B) correspond to mean ± SD of four independent experiments (***P < 0.001, ns – not statistically significant). **(C)** The model summarizing the results shown here: like the yeast Ypt1, both Rab1A and Rab1B control the ER-to-Golgi step of the secretory pathway (top), whereas only Rab1A, and not Rab1B, regulates AP formation in stress-induced autophagy (bottom).

level of the AP membrane available in Rab1AKO and Atg9KO cells to which LC3II can attach would lead to a reduction in LC3II level (either due to inhibition of LC3 lipidation or instability of LC3II when not in membranes).

### Mechanism of Rab1A specificity in autophagy

We used the fact that Rab1B does not function in autophagy to create a chimera in which the HVD of Rab1A (the 25 C-terminal amino acids) replaces the HVD of Rab1B. This chimera behaves like Rab1A in complementing the autophagy defect of Rab1AKO cells, AP formation, and the localization to APs. Thus, the HVD of Rab1A is sufficient to render a Rab1B that can function in autophagy and localize to APs.

### Importance of these findings

The importance of these results is twofold: in providing a mechanism for dual activity of a single Rab GTPase in two different pathways and for human health. First, to our knowledge, this is the first demonstration of a role for the HVD domain in granting a single Rab a dual role in two completely different trafficking pathways: secretion and autophagy. Although the importance of HVD for the localization of different Rab subfamilies to specific organelles is known (Chavrier et al, 1991; Li et al, 2014), we show a differentiating role for the HVD in the cellular function of two nearly identical Rabs from the same subfamily. Rab1A and Rab1B play similar functions in one pathway, that is, secretion, and HVD enables Rab1A to function in a different pathway, that is, autophagy. Many of the ~70 human Rabs have multiple paralogs (Homma et al, 2021), and HVD-specificity could play a role in other processes too. Second, Rab1A and Rab1B were implicated in multiple human diseases that range from cancer to neurodegeneration (Winslow et al, 2010; Coune et al, 2011; Thomas et al, 2014; Xu et al, 2015; Halberg et al, 2016). The two processes they regulate were also shown to be important for cancer (Yun & Lee, 2018; Del Giudice et al, 2022) and neurological disorders (Nah et al, 2015; Kuo et al, 2021). Data presented here enable teasing apart the association of Rab1A or Rab1B with these diseases (see below).

### Future questions

Findings presented here raise new mechanistic questions about the role of Rab1A/B in other types of autophagy, the exact mechanism by which Rab1A-HVD functions in autophagy versus secretion, the possibility that HVD plays a similar role in other Rabs, and the idea that Rabs can coordinate different pathways. First, although we show here that only Rab1A, and not Rab1B, plays a role in stress-induced autophagy, the question of which Rab1 plays a role in selective autophagy processes is still open. For example, mitophagy and granulophagy, which clear damaged mitochondria and protein aggregates, respectively, are important for cancer and neurodegenerative diseases (Frankel et al, 2017; Killackey et al, 2020). It is still an open question whether Rab1A and/or Rab1B play roles in such processes. Second, it is important to determine what in Rab1A-HVD is key for its role in stress-induced autophagy, for example, which of the eight amino acids that differ between

Rab1A and Rab1B are crucial for differentiating Rab1A and Rab1B, and if post-translation modifications and/or interactions with other proteins, for example, guanine-nucleotide exchange factor, GDI or effectors, are involved in its dual functionality. Third, Rab1A is the first example of Rab that can coordinate secretion and early autophagy. Rab5 is another example of a Rab that can coordinate endocytosis and late autophagy (Chen et al, 2014; Zhou et al, 2017). It is interesting to explore whether HVD domains of other human Rab subfamilies can confer specific functions, for example, Rab3 A-D, Rab5 A-C, and Rab6 A-C (Homma et al, 2021). Finally, we have proposed that Rab GTPases play roles not only in regulation and coordination of individual transport steps in a pathway, but also in coordination of different pathways (Lipatova et al, 2015). While coordination between pathways is logical for the optimal functioning of cells, there is currently no evidence that it exists. The finding that Rab1A is required for two different pathways provides means for testing this idea.

Future questions about Rab1A/B in human health would center around their association with acquired diseases associated with aging, that is, cancer and neurological disorders. Because activation of both Rab1A and Rab1B were implicated in multiple cancer types (Yang et al, 2016), both should be considered when studying cancer. In contrast, only Rab1A, and not Rab1B, has been associated with neurological disorders, for example, Parkinson's disease (Winslow et al, 2010; Coune et al, 2011; Ejlerskov et al, 2013; Mazzulli et al, 2016; Hatstat et al, 2022), Alzheimer's disease (Mohamed et al, 2017), amyotrophic lateral sclerosis (Webster et al, 2018), and intellectual disability (Tabata et al, 2022). In light of the results presented here, we propose that it is the function of Rab1A in autophagy that is important for its role in neurological disorders. The finding that Rab1A can be inactivated whereas Rab1B provides the essential function in secretion and cell viability would allow studying the importance of secretion versus autophagy in these diseases. Moreover, the ability to target Rab1A without affecting cell growth and viability is crucial for targeted therapeutic design.

## Materials and Methods

### Cell lines and plasmids

HEK293T cells, WT, and Rab1BKO were a kind gift from Dr. Stacy Horner (Duke University) (Beachboard et al, 2019). HAP1 cells, WT, and Rab1AKO were a kind gift from Dr. Thijn Brummelkamp (The Netherlands Cancer Institute) (Blomen et al, 2015). Rab1AKO in HEK293T cells, Rab1BKO in HAP1 cells, and Atg12KO in HEK293T and HAP1 cells were generated in our laboratory. The plasmids used in this study are RAB1A cDNA ORF Clone, Human, N-Myc tag (HG16400-NM; SinoBiological), RAB1B cDNA ORF Clone, Human, N-HA tag (HG15447-NY; SinoBiological), pEGFP-C1-mouse Rab1A, constructed by Mitsunori Fukuda (2004.1.14), pEGFP-C1-mouse Rab1B, constructed by Mitsunori Fukuda (2004.1.31) (Ishida et al, 2012), pNL1.3CMV(SecNluc) plasmid-encoding NanoLuciferase was a kind gift from Prof. Paul Melancon. The oligonucleotides were bought from Integrated DNA Technologies.

We generated the Rab1A gene KO in HEK293T cells using CRISPR/Cas9, according to jetCRISPR RNP transfection reagent protocol for reverse transfection. In brief, an RNP complex was formed by incubation of gRNA with Cas9. The complex was transfected into HEK293T WT cells using lipid nanoparticles which resulted in gene editing at the genomic target site. DNA from single-cell clones with the desired edit was isolated and sequenced to confirm the KO (Elkhadragy et al, 2021).

The Rab1BA chimera plasmid was constructed by homologous re-combinational cloning (Jacobus & Gross, 2015). Specifically, one pair of primers were designed to amplify the AA 1–AA 177 of the N terminus of Rab1B from pEGFP-C1-Rab1B plasmid and about 4.5-kb part of the pEGFP-C1 vector before the Rab1B gene. pEGFP-C1-Fwd: 5′-GCTCATGAGACAATAACCCTGATAAATGCTTC-3′ and Rab1B-Rev: 5′-TGCTCCTGGCCCCATCCG-3′. Another pair of primers to amplify the AA 181–AA 205 of C terminus Rab1A and about 800 bp of the rest of the vector sequence after the Rab1A stop codon sequence from pEGFP-C1-Rab1A plasmid.Rab1A-Fwd: 5′-GCTGCAGAGATCAAAAAGCG-GATGGGGCCAGGAGCAACAGCTGGTGGTGCCGAG–3′. pEGFP-C1-Rev: 5′-GAAGCATTTATCAGGGTTATTGTCTCATGAGC-3′. Rab1A-Fwd and pEGFP-C1-Rev have their N terminus homologous with Rab1B-Rev or pEGFP-C1-Fwd. The two PCR products were then co-transformed into NEB 5-α F′Iq Competent *Escherichia coli* (C2992I; New England BioLabs) and recombinant plasmids resulting from in vivo homologous recombination were then isolated, purified, and confirmed by sequencing. The obtained plasmids were then used to transfect HEK293T cells.

## Reagents

The following antibodies were used in this study: rabbit Anti-LC3B (2775S; Cell Signaling Technology), for immunofluorescence and Western blotting, Anti-SQSTM1/p62 (2C11) (ab56416; Abcam) for immunofluorescence and Western blotting, rabbit Anti-LAMP1 (D2D11) XP, (9091; Cell Signaling Technology) for immunofluorescence, mouse Anti-LAMP1 (D4O1S) (15665; Cell Signaling Technology), Anti-Atg12 (Human Specific) (2010S; Cell Signaling Technology), mouse monoclonal Anti-Rab1B (SAB1400720; Sigma-Aldrich), rabbit polyclonal Anti-Rab1A (11671-1-AP; Proteintech), rabbit monoclonal Anti-GM130 (D6B1) (12480; Cell Signaling Technology), mouse monoclonal anti–β-actin (A5441; Sigma-Aldrich), mouse monoclonal Anti-Myc/c-Myc Antibody (9E10; Santa Cruz), mouse Anti-GFP (11814460001; Roche). The HRP-conjugated secondary antibodies were purchased from Cytiva, ECL Mouse IgG, HRP-linked whole Ab from sheep (NA931V) and ECL rabbit IgG, HRP-linked whole Ab from a donkey (NA934V). All Alexa-conjugated secondary antibodies were purchased from Jackson ImmunoResearch Laboratories, Fluorescein (FITC) AffiniPure Goat Anti-Rabbit IgG 111-095-144, Alexa Fluor 594 AffiniPure Donkey Anti-Mouse IgG 715-585-150, and Alexa Fluor 647 AffiniPure Goat Anti-Rabbit IgG 111-605-144. All the media were purchased from Sigma-Aldrich.

## Cell culture

HEK293T cells were cultured in DMEM supplemented with 10% vol/vol FBS, 100 U/ml penicillin–streptomycin. HAP1 cells were cultured in IMDM media supplemented with 10% vol/vol fetal bovine serum;

100 U/ml penicillin-streptomycin as recommended by Horizon Discovery. Cells were maintained in an incubator with 37°C, 5% $CO_2$, and humidified atmosphere, and passaged approximately every 3 d using Trypsin-EDTA solution (Sigma-Aldrich).

## Transfection

For transient transfections, HEK293T cells were either seeded on glass coverslips on 24-well plates or directly on six-well plates. When cells were 60–70% confluent, transfections were performed with Lipofectamine 2000 (Invitrogen), according to the manufacturer's instructions. Briefly, a mixture of optiMEM with 1 μg of DNA was incubated at room temperature for 5 min. Another mixture containing optiMEM and Lipofectamine 2000 was prepared and incubated for 5 min. Both solutions were then mixed and incubated for ~20 min. The mixture was then added to the cells. For stable expression, HEK293T and HAP1 cells were transfected with Lipofectamine 2000 (Invitrogen), according to the manufacturer's instructions, following the same protocol as mentioned previously and subsequently, cells were selected with 0.7 mg/ml hygromycin.

## Immunofluorescence microscopy

For immunofluorescence, cells were grown on poly-L-lysin–coated coverslips overnight to 60–80% confluence. After incubation under different experimental conditions, cells were fixed with ice-cold methanol: acetone (2:1) for 5–10 min at −20°C or with 4% PFA followed by permeabilization with 0.25% Triton X-100. Cells were washed with PBS and blocked with 2% BSA and 5% goat serum for 1 h at room temperature to reduce the non-specific binding of the primary and secondary antibodies. Cells were then incubated with primary antibodies at appropriate dilution for 1 h at room temperature or overnight at 4°C. Coverslips were then washed three times with PBS and incubated with secondary antibodies for 1 h at room temperature. The primary and secondary antibodies were prepared in the blocking buffer. The dilution of the secondary antibodies used here was 1:800. After three washes with PBS, the coverslips were mounted on glass slides with ProLong Diamond Antifade Mountant (Invitrogen). The coverslips were then imaged using Zeiss LSM 700 laser scanning confocal microscope or Yokogawa Spinning Disk confocal Leica DM8i inverted microscope.

## Luciferase secretion assay

HEK293T cells were transfected with pNL1.3CMV(SecNIuc) plasmid-encoding NanoLuciferase. For secretion measurements, cells were washed with serum-free medium placed in a fresh medium with or without BFA (10 μg/ml for 2 h). The luciferase assay was done as previously described (Kumar et al, 2016). At the indicated time points 0, 20, 40, 60, 80, 100, and 120 min, the medium was collected and total luciferase in cell lysates and it was assessed using the luciferase substrate coelenterazine prepared at 1.4 μm in a luciferase assay buffer (25 mm glycylglycine, pH 7.8, 15 mm $K_2PO_4$, pH 7.8, 15 mm $MgSO_4$, and 4 mm EGTA). Total cell luciferase was measured after lysing the cells in 200-μl luciferase lysis buffer (0.1% [vol/vol] Triton X-100, 25 mM glycylglycine, pH 7.8, 15 mM $MgSO_4$, 4 mM EGTA, and 1 mM dithiothreitol). The signal was quantitated with a

fluorescence microplate reader. A fraction of the secreted luciferase was calculated as (total signal from growth medium)/(total signal from growth medium + total signal from lysed cells).

## Autophagy analysis by fluorescence microscopy

Autophagy in HEK293T and HAP1 cells was induced by serum and amino acid deprivation or by rapamycin treatment (200 nM). Cells were cultured in complete media for 24 h, then washed and incubated either in EBSS or with rapamycin for 3 h in the presence or absence of 100 nM bafilomycin A1 (BafA1) during the last 1.5 h. Autophagy was determined by counting the LC3 puncta per cell, the LC3/p62 colocalized dots (likely corresponding to APs), and the LC3/LAMP colocalizing puncta. For quantification, 5–10 randomly selected fields per slide representing about 80 cells per data point were taken at 63× objective. The number of puncta was automatically counted using semi-automated counting in Adobe Photoshop using raw images. Threshold was adjusted to highlight all the structures to be counted. After determining the threshold, the background noise was removed. The same parameters were applied to all the images to be quantified. For each single fluorescent image, the total number of puncta present was determined, and the average amount in each images was calculated. The number of cells and/or colocalizing foci was manually recorded using the counting tool in Photoshop. Number of cells was determined by counting DAPI-stained nuclei (this number was corrected for partial cells in the image edges). Colocalization was determined by counting puncta that do or do not overlap on a single plane. To avoid variations, all quantifications were carried out by the same person. For statistical analyses, we used $t$ test with mean ± SD (*$P < 0.05$, **$P < 0.01$, ***$P < 0.001$, ****$P < 0.0001$, $ns$ – not statistically significant) in GraphPadPrism software. Panels in the figures represent cropped fields of microscopy images and were processed using Adobe Photoshop. Specifically, for setting the brightness, contrast, sharpness, and background removal, the untreated WT cells were used as a standard and these settings were applied to all the panels in the experiment.

## Western blot analysis

HEK293T and HAP1 cells were seeded on a six-well plate and, following treatment cells, were washed twice in ice-cold PBS containing protease and phosphatase inhibitors (0.5 mM PMSF, EDTA-free protease inhibitor cocktail tablet). Cells were then harvested and lysed with an NP-40–based lysis buffer (150 mMNaCl, 50 mM Tris–HCl, pH 7.4, 1 mM EDTA, and 1% NP-40) (Sharifi et al, 2015). The samples were denatured in Laemmli's buffer and the proteins were resolved on 12% SDS–PAGE gel and transferred to a PVDF membrane overnight at 4°C. The protein bands were detected after incubation with the appropriate primary antibodies for 1–2 h at room temperature, except incubation with anti-Rab1A and anti-Rab1B which was done overnight at 4°C. After 3× washing with Tris-buffered saline with Tween-20 (50 mm NaCl, 0.5% [vol/vol] Tween-20, 20 mm Tris-HCl, pH 7.5), the HRP-conjugated secondary antibodies were applied. The protein bands were visualized by ECL (GE Healthcare) and exposed to X-ray films. Quantification was done using ImageJ or Image Studio Lite software (LI-COR

Biotechnology). For Image Studio Lite, a tiff image from the Western blot was scanned from an x-ray film. The bands to be analyzed were manually selected and a rectangle was drawn to encompass the band in the first lane, large enough to enclose each of the remaining bands. An identically sized box was automatically added to the subsequent lanes for the background noise to be removed from all of the samples. Overexposed bands with a value of infinity were excluded from the quantification. The raw signal values were exported to an Excel spreadsheet and the relative density of the samples were calculated and normalized to the first lane, used as a standard. For statistical analyses, we used $t$ test with mean ± SD (*$P < 0.05$, **$P < 0.01$, ***$P < 0.001$, ****$P < 0.0001$, $ns$ – not statistically significant) in GraphPadPrism software.

# Supplementary Information

# Acknowledgements

We thank TR Brumelkamp (The Netherland Cancer Institute, Amsterdam, NL) and S Horner (Duke University, Durham, NC), for providing us with WT and KO cell lines; M Regan, B Merrill, and K Hodges, for help with generation of KO cell lines; P Melancon and C Chan (University of Alberta, Edmonton, CA), for sending us the plasmid and protocol for the luciferase secretion assay; M Fukuda (Tohoku University, Sendai, JP) for the GFP-Rab1A and GFP-Rab1B; and N Hay for advice. This research was funded by grants RO1GM045444 and R35GM141479 from GM, and R21NS099556 from NINDS to N Segev.

## Author Contributions

V Gyurkovska: conceptualization, data curation, investigation, visualization, methodology, and writing—original draft, review, and editing.
R Murtazina: investigation and visualization.
SF Zhao: data curation and investigation.
S Shikano: investigation.
Y Okamoto: investigation.
N Segev: conceptualization, resources, data curation, formal analysis, funding acquisition, investigation, methodology, project administration, and writing—original draft, review, and editing.

## Conflict of Interest Statement

The authors declare that they have no conflict of interest.

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
