## [Reviewer comments · Life Science Alliance]

Dual function of Rab1A in secretion and autophagy: hyper-variable domain dependence

Valeriya Gyurkovska, Rakhilya Murtazina, Sarah F. Zhao, Sojin Shikano, Yukari Okamoto and Nava Segev

DOI: 10.26508/lsa.202201810

Corresponding author(s): Prof. Nava Segev (University of Illinois at Chicago)

Review timeline:

Submission Date:	2022-11-08
Editorial Decision:	2022-11-10
Revision Received:	2023-01-12
Editorial Decision:	2023-02-01
Revision Received:	2023-02-01
Accepted:	2023-02-02

Scientific Editor: Eric Sawey

Transaction Report:

No Peer Review Process File is available with this article, as the authors have chosen not to make the review process public in this case.

Re: Life Science Alliance manuscript #LSA-2022-01810-T

Dr Nava Segev
University of Illinois at Chicago
Dept of Biochemistry and Molecular Genetics
Molecular Biology Research Building
Rm. 2252
900 South Ashland Avenue, Chicago IL 60607

Dear Dr. Segev,

Thank you for submitting your manuscript entitled "Dual function of Rab1A in secretion and autophagy: hyper-variable domain dependence" to Life Science Alliance. We invite you to submit a revised manuscript addressing the following Reviewer comments:

- Address Reviewer 1's Major Points #3-5
- Address Reviewer 2's comments.
- Address Reviewer 3's Specific Comments.

Thank you for this interesting contribution to Life Science Alliance. We are looking forward to receiving your revised manuscript.

Sincerely,

- A letter addressing the reviewers' comments point by point.

B. MANUSCRIPT ORGANIZATION AND FORMATTING:

RE: Life Science Alliance Manuscript #LSA-2022-01810-TR

Prof. Nava Segev
University of Illinois at Chicago
Dept of Biochemistry and Molecular Genetics
Molecular Biology Research Building
Rm. 2252
900 South Ashland Avenue, Chicago IL 60607

Dear Dr. Segev,

Thank you for submitting your revised manuscript entitled "Dual function of Rab1A in secretion and autophagy: hyper-variable domain dependence". We would be happy to publish your paper in Life Science Alliance pending final revisions necessary to meet our formatting guidelines.

- please address Reviewer 3's remaining comment
- please add ORCID ID for corresponding author-you should have received instructions on how to do so
- please upload your main and supplementary figures as single files
- please make sure that the author list in the manuscript and our system match and that all authors are added to the system

Figure Check:

- please add scale bars to Figure S5A and Figure S6D

A. FINAL FILES:

-- High-resolution figure, supplementary figure and video files uploaded as individual

files: See our detailed guidelines for preparing your production-ready images, <https://www.life-science-alliance.org/authors>

B. MANUSCRIPT ORGANIZATION AND FORMATTING:

Sincerely,

3rd Editorial Decision

02 February 2023

RE: Life Science Alliance Manuscript #LSA-2022-01810-TRR

Prof. Nava Segev
University of Illinois at Chicago
Dept of Biochemistry and Molecular Genetics
Molecular Biology Research Building
Rm. 2252
900 South Ashland Avenue, Chicago IL 60607

Dear Dr. Segev,

Thank you for submitting your Research Article entitled "Dual function of Rab1A in secretion and autophagy: hyper-variable domain dependence". It is a pleasure to let you know that your manuscript is now accepted for publication in Life Science Alliance. Congratulations on this interesting work.

DISTRIBUTION OF MATERIALS:

Again, congratulations on a very nice paper. I hope you found the review process to be constructive and are pleased with how the manuscript was handled editorially. We look forward to future exciting submissions from your lab.

Sincerely,
